# Pilot study on the effect of a Meditation–Mindfulness–Positive Psychology Training program on perceived stress and mental well-being in Korean nursing students: A mixed methods analysis

Young Im Cho[1⊘], Hyo Jin Kim[iD][2⊘], Dong Hee Kim[iD][3]*

**1** Department of Psychology, Duksung Women's University, Dobong-gu, Seoul, Republic of Korea,
**2** College of Nursing, Seoul National University, Jongno-gu, Seoul, Republic of Korea, **3** College of Nursing, Sungshin Women's University, Gangbuk-gu, Seoul, Republic of Korea

⊘ These authors contributed equally to this work.
* dhkim@sungshin.ac.kr

## Abstract

Nursing students experience stress that negatively affects their mental well-being and academic performance. Conventional mindfulness-based interventions are too long to fit the students' demanding schedules. This study aimed to examine the feasibility and preliminary effectiveness of an adapted Meditation–Mindfulness–Positive Psychology Training (MMPT) program tailored to nursing students' needs. The program was designed as to be shorter, to be conducted online, and to be tailored to the needs of nursing students. A mixed-methods pilot study was conducted among junior and senior nursing students in Korea. The experimental group participated in a six-session online MMPT program into which meditation, mindfulness, gratitude journaling, and self-compassion practice were incorporated. Quantitative outcomes were assessed before and after intervention. Qualitative data were collected through in-depth interviews and analyzed through inductive content analysis. The quantitative results showed significant improvements in gratitude disposition, self-compassion, and perceived stress, but not in mindfulness or overall mental health. Qualitative findings complemented these results by revealing perceived improvements across all target domains, with participants reporting enhanced calmness, greater self-kindness, increased positivity, and stronger coping strategies. Students highlighted the value of gratitude journaling, compassion-based exercises, and supportive peer interactions, which reinforced their engagement and sense of belonging. The MMPT program adapted for this study demonstrated feasibility and preliminary effectiveness in reducing stress and enhancing positive psychological resources among nursing students. By integrating mindfulness and positive psychology into a brief and accessible format, this study underscores the theoretical and practical value of needs-based

**Data availability statement:** All relevant data are within the paper and its Supporting Information file.

**Funding:** This work was supported by the Sungshin Women's University Research Grant 2025 (grant number H20250071) awarded to KDH. The funder was Sungshin Women's University (https://www.sungshin.ac.kr). The funder had no role in study design, data collection and analysis, decision to publish, or preparation of the manuscript.

**Competing interests:** The authors have declared that no competing interests exist.

interventions for supporting nursing students' well-being. Further large-scale and longitudinal studies are warranted to validate and extend these findings.

## Introduction

Nursing students, particularly those in their junior and senior years, experience high levels of stress owing to demanding academic workloads, prolonged clinical practicums, and the pressure to pass national licensure examinations [1]. Studies have shown that nursing students, both in Korea and internationally, report significantly higher levels of stress, anxiety, and depressive symptoms than students in other academic disciplines [2,3]. Such sustained academic and clinical stress has been associated with emotional exhaustion, burnout, and compromised mental well-being [4,5], which may ultimately interfere with students' learning, professional development, and future clinical performance [5–7].

Early stress management education during undergraduate training is essential for mitigating these negative outcomes. Despite this need, most nursing programs in Korea do not provide structured training on stress management, emotional regulation, and self-care. Consequently, nursing students often resort to informal coping strategies that may prove ineffective or unsustainable [1,8]. Studies indicate that mindfulness-based techniques can improve resilience and promote psychological well-being [9,10]. Students who have received such training have reported greater satisfaction with their majors, lower academic burnout, and increased readiness for clinical practice [11].

Despite evidence supporting the effectiveness of mindfulness-based interventions for stress management, several practical and theoretical limitations persist. Time is a significant challenge in implementing mindfulness-based stress management programs. For example, nursing students often face constraints from full academic schedules and long clinical hours, which make it difficult to participate in conventional mindfulness-based interventions, such as the standard 8-week Mindfulness-Based Stress Reduction program [12,13]. Moreover, mindfulness-only interventions have theoretical and experiential limitations. While mindfulness promotes nonjudgmental awareness and emotional regulation, it may not sufficiently cultivate positive emotional states, intrinsic motivation, or resilience [14,15]. Kim [16] noted that mindfulness in isolation may lead to emotional disengagement or passivity and emphasized the importance of integrating mindfulness with positive psychological interventions, such as gratitude, strength-based reflection, and optimism training, to actively build psychological resources and well-being.

Consistent with this perspective, research on positive psychotherapies has demonstrated that self-affirmation and self-compassion function as key intrapersonal mechanisms that enhance self-worth, emotional regulation, and perceived mental well-being [17–19]. These findings provide a theoretical foundation for integrating mindfulness with positive psychology to address stress through not only awareness but also adaptive cognitive and emotional restructuring.

An integrated approach of combining mindfulness with positive psychology has been proposed as a more comprehensive method for reducing stress and promoting flourishing. While mindfulness enhances self-awareness and acceptance, positive psychology fosters emotional growth and resilience through practices for building positive affect, such as gratitude and self-compassion [16]. Research indicates that combining MBIs with positive psychology interventions yields synergistic effects that surpass the benefits of one of these components alone [20]. In Korea, Kim [2,16,21] developed the Meditation–Mindfulness–Positive Psychology Training (MMPT) program, in which these two approaches have integrated into a structured format. The MMPT program was developed with the aim of enhancing participants' emotional resilience and well-being by balancing mindful awareness with strength-based positive emotion cultivation [22].

Empirical studies have demonstrated the effectiveness of MMPT programs in reducing stress and burnout and improving emotional well-being and life satisfaction across various populations, including students, teachers, and community members [2,22,23]. However, evidence on brief, needs-based MMPT adaptations for nursing students—particularly those delivered online—remains limited. This gap highlights the need for tailored interventions that integrate mindfulness with positive psychological mechanisms while addressing the practical constraints faced by nursing students.

Given these considerations, the present study aimed to develop and evaluate a stress management program based on the MMPT framework, adapting it to the specific needs and time constraints of third- and fourth-year nursing students in Korea. A convergent mixed-method design was employed in the study. Quantitatively, the study aimed to assess whether students who participated in the MMPT program demonstrated higher levels of mindfulness, gratitude disposition, and self-compassion, along with lower perceived stress and improved mental well-being than the control group. Qualitatively, this study explored students' experiences and perceived changes following the program.

## Methods

### Participants and procedure

This study included junior and senior nursing students who enrolled in baccalaureate programs at a university in Seoul, South Korea. After the Sungshin Women's University Institutional Review Board (SSWUIRB-2023–025, 050) approved the study, participants were recruited through structured announcements posted in an online nursing student community at two universities. Before commencing the study, the researcher explained to the participants the purpose and methodology of the study via an online information sheet in detail. Students who voluntarily agreed to participate provided their consent through an online consent form. The participants were informed that they could withdraw from the study at any time without penalty, and that their identity and confidentiality would be strictly protected, as the collected data would be used solely for research purposes. The surveys and interviews did not include any personal identifiers, and the authors did not have access to information that could identify individual participants during or after data collection.

The experimental group comprised 16 participants. Of these, 13 completed pre- and post-test assessments. The pre-test was conducted before the program, and the post-test was completed within one week after the program. One month after the completion of the program, in-depth interviews were conducted to explore participants' experiences and perceived effects of the program. Three participants were excluded from the final analysis owing to early withdrawal or repeated absences (more than twice) attributed to employment, hospitalization, or academic pressure. From February 2 to March 6, 2024, the program was conducted weekly for six weeks, when the participants divided into two groups met on different weekdays. The participants who could not attend their designated sessions were allowed to join the sessions of other groups in the same week. For those who missed a session, a video recording of the session was provided to ensure that they could compensate for the missing session. All recorded sessions followed the same standardized structure, content, and practice assignments as the live sessions to minimize potential variation in intervention delivery. Each session included assigned tasks, and the participants were instructed to track their completion through a shared online file. The program leader monitored the participants' task completion and engagement. Open-group chat was used to share audio

files and session summaries, encourage participation, and provide a platform for questions and answers. The authors conducted all program activities and facilitations.

The participants of the control group were recruited online from a university with a similar size, and the curriculum like that of the experimental group was employed. Twelve students completed the pre-test in February and the post-test in March 2024— six weeks apart. (Fig 1,2)

To minimize potential experimenter bias, the intervention followed a standardized protocol, and quantitative assessments were collected using validated self-report instruments with predefined scoring procedures.

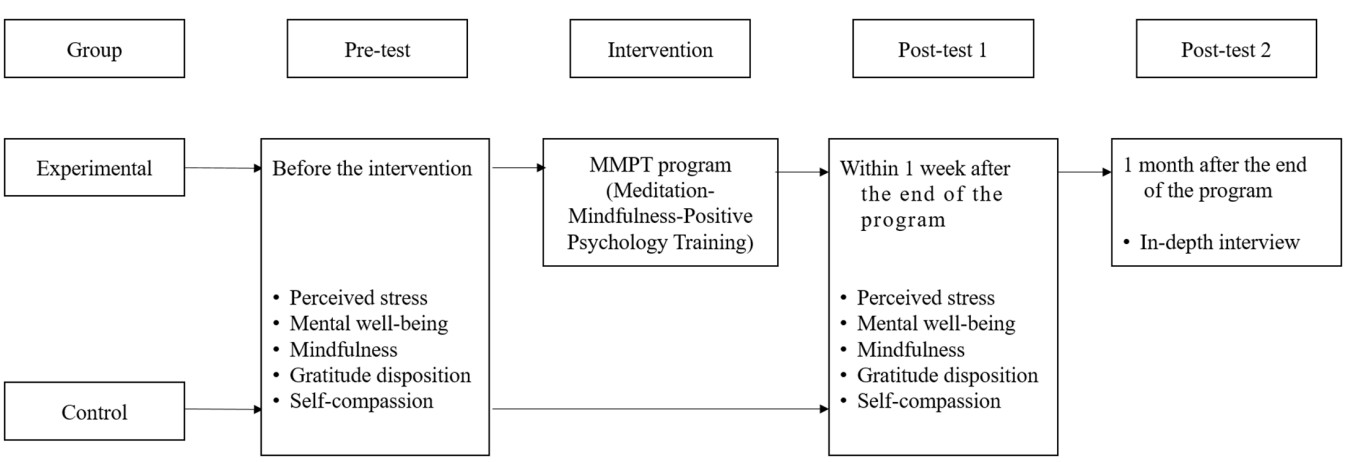

**Fig 1. Study design.**

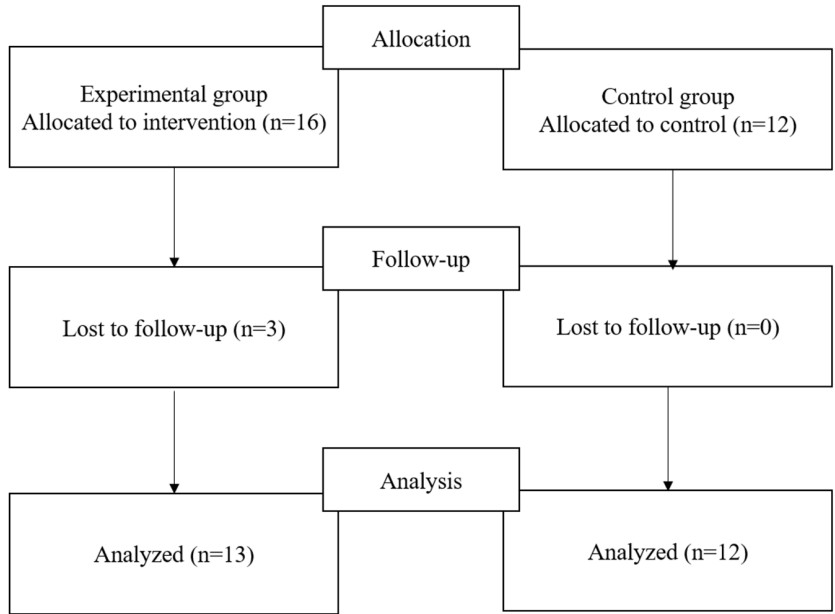

**Fig 2. Flow diagram of participants' enrollment.**

## Intervention

In this study, a modified version of the MMPT program originally developed by Kim was used as an intervention [24]. The original program, in which mindfulness and positive psychology had been integrated, was adapted to suit the context of nursing students. First, a needs assessment was conducted with two nursing faculty members and six nursing students (see Table 1). The author conducted the assessment through individual online interviews lasting 30–60 min. The results indicated that all components of the original MMPT program were relevant. However, the needs assessment revealed that nursing students faced substantial time constraints due to academic workload and clinical training, as well as practical limitations related to in-person participation. Accordingly, the program was adapted by reducing the number of sessions from eight to six and shortening each session from 2 hours to 90 minutes. To enhance accessibility and feasibility, the intervention program was delivered in an online format (see Table 1). Based on the original program and the results of the needs assessment, a modified program was developed and finalized after the original developer of the MMPT validated the content (Table 2).

## Variables and instruments

**Demographic characteristics.** Demographic variables included sex, age, academic year, religion, perceived health status, and engagement in regular exercises.

**Perceived stress.** The Perceived Stress Scale (PSS), originally developed by Cohen et al. [25], was later shortened through factor analysis by Cohen and Williamson [26]. This abbreviated version was translated into Korean and validated with university students by Park and Seo [27]. The scale consists of 10 items rated on a 5-point Likert scale and comprises two factors: negative and positive perceptions.

**Mental well-being.** The Korean version of the Mental Health Continuum Short Form (K-MHC-SF) [28], validated in Korea based on the original MHC-SF [29] was used to assess mental well-being. The scale consists of 14 items rated on a six-point Likert scale, ranging from "every day," "almost every day," "approximately once a week," "approximately 2–3 times a week," "once or twice," to "never." It is used to measure three dimensions of wellbeing: psychological, social, and

**Table 1. Components of MMPT Program based on Kim [21] and needs assessment.**

|  | Needs assessment | Application of the program in this study based on Kim [21] |
|---|---|---|
| Program content | Recognizing, accepting, and expressing needs, emotions, states, strengths, and limitations objectively; | Meditation, Mindfulness |
|  | desire to learn how to pause excessive thoughts; need for relaxation techniques or meditation that can be practiced during busy moments |  |
|  | Need to identify a plan and engage in enjoyable activities independently. | Positive Psychology Well-being behaviors |
|  | Developing and utilizing self-directed messages. |  |
|  | Need to acknowledge oneself and cultivate self-compassion | Positive Psychology Gratitude and compassion |
| Time constraints | High pressure owing to concurrent academic work and clinical practice; lack of personal time | Reduction in number and duration of program sessions Developing and adapting online programs |
| Methods | Need for increased time in direct interpersonal interactions | Facilitating peer communication through group chat platforms |

**Table 2. MMPT Program for nursing students.**

| Session | Title | Content summary |
|---|---|---|
| 1st | Becoming familiar with one's senses | Introduction of the teacher and participants, overview of group agreements and the MMPT program.<br>Sharing personal stress experiences and psychoeducation on stress.<br>Practice: Five-sense meditation and activity meditation.<br>Sharing of experiences and Q&A.<br>Homework assigned. |
| 2nd | Becoming familiar with one's body | Practice: Activity meditation.<br>Sharing of reflections and feedback on the past week's practice.<br>Practice: Yoga meditation, body meditation, and breathing meditation.<br>Sharing of experiences, Q&A, and feedback.<br>Homework assigned |
| 3rd | Mindfulness in daily life | Practice: Breathing meditation.<br>Sharing of reflections and feedback on the past week's practice.<br>Explanation of the distinction between meditation and mindfulness.<br>Introduction to the concept of Zero-I.<br>Practice: Breath-based mindfulness meditation.<br>Sharing of experiences, Q&A, and feedback.<br>Psychoeducation on emotional mindfulness.<br>Practice: mindfulness of stress.<br>Sharing of experiences, Q&A, and feedback.<br>Homework assigned |
| 4th | Positive psychology: Well-being behavior & well-being cognition | Practice: Mindfulness of well-being.<br>Sharing of reflections and feedback on the past week's practice.<br>Psychoeducation on Mind Society Theory and Positive Psychology.<br>Practice: Well-being behavior.<br>Practice: Well-being cognition.<br>Sharing of experiences, Q&A, and feedback.<br>Homework assigned. |
| 5th | Positive psychology: Gratitude and compassion | Practice: Breath-based mindfulness meditation.<br>Sharing of reflections and feedback on the past week's practice.<br>Practice of gratitude and writing a well-being journal.<br>Practice: Compassion-based exercises.<br>Sharing of experiences, Q&A, and feedback.<br>Homework assigned. |
| 6th | Review of the entire program & planning for continued practice | Practice: Yoga-based mindfulness meditation and 3-minute breath-based mindfulness meditation.<br>Sharing of reflections and feedback on the past week's practice.<br>Practice and sharing: The Compassionate Chair and 'What Do You Want?'<br>Summary of the entire program.<br>Sharing of reflections on the overall experience.<br>Writing a plan for continued personal practice.<br>Closing with a compassion-based exercise. |

emotional. Higher scores indicated greater perceived mental well-being. In this study, the scale demonstrated high internal consistency, with a Cronbach's alpha of.89.

**Mindfulness.** Mindfulness was assessed with the Korean version of the Five-Facet Mindfulness Questionnaire Short Form, validated by Cheong et al. [30]. This instrument is an abbreviated and validated version of the Korean adaptation developed by Won and Kim [31]. The original version was based on the original Five-Facet Mindfulness Questionnaire developed by Baer et al. [32]. The scale employed in this study consisted of 15 items categorized into five domains: non-reactivity, observing, acting with awareness, describing, and non-judging of experience. Each item was rated on a seven-point Likert scale, with higher scores indicating greater mindfulness. In this study, the scale demonstrated acceptable internal consistency, with a Cronbach's alpha of.76.

**Gratitude disposition.** Gratitude disposition was assessed using the Gratitude Questionnaire (GQ-6) developed by McCullough et al. [33]. The validated Korean version of the Gratitude Disposition Questionnaire (K-GQ-6) by Kwon and Kim was used in the study [34]. This scale consists of six items rated on a seven-point Likert scale, with higher scores indicating a stronger gratitude disposition. In this study, the scale demonstrated high internal consistency, with a Cronbach's alpha of.87.

**Self-compassion.** Self-compassion was assessed with the Korean version of the Self-Compassion Scale Short Form (K-SCS-SF), validated by Kim et al. [35] based on the original SCS-SF by Raes et al. [36]. The scale consists of 12 items categorized into six dimensions rated on a five-point Likert scale, with higher scores indicating greater self-compassion. The scale demonstrated acceptable internal consistency with a Cronbach's alpha of.75.

**Qualitative measurement.** In-depth interviews were conducted to explore participants' experiences with the MMPT program. The primary interview question was "How was your experience participating in the MMPT program, and how has it influenced your life?" The follow-up questions were used to assess their response in several areas. The participants were asked about their reasons or motivations for participating in the program. They were also invited to describe any personal changes that they experienced as a result of the program, including those in their perceptions and responses to stress. Additional questions were used to explore the program's impact on their clinical practicum, any difficulties or positive aspects that the participants encountered during participation, and how they had applied the content of the program after its completion. Finally, the participants were asked whether they had discovered anything new about themselves, experienced changes in self-perception, or noticed any changes in their relationships with themselves, professors, peers, family members, or patients.

## Data analysis

Data were analyzed with the SPSS software (version 25.0) and R with ARTool [37] emmeans package for Aligned Rank Transform analysis of variance (ART ANOVA) [38]. Percentages, means, and standard deviations were calculated to summarize participants' general characteristics. Given the small sample size of 25, a nonparametric mixed-design analysis of variance was conducted to examine changes across groups and time points. To this end, an Aligned Rank Transform (ART) method was employed. ART ANOVA is a nonparametric approach for maintaining the design and interpretive structure of traditional ANOVA without requiring the assumptions of normality and homogeneity of variance. This is particularly advantageous in factorial designs, as it facilitates the analysis of main effects, including interaction effects, which are typically not accessible through conventional rank-based tests. Unlike simpler nonparametric tests, ART ANOVA enables simultaneous examination of group, time, and interaction effects, which was essential for the repeated-measures design utilized in this study. ART ANOVA is used to align and rank data of each effect separately and fit a linear model. Then the analysis is performed with traditional ANOVA [39]. To verify the simple main effect of the group factor following the ART ANOVA, we additionally conducted Wilcoxon signed-rank tests.

Interview data were analyzed with inductive content analysis following the method of Elo and Kyngäs [40]. The qualitative analysis was conducted by researchers with prior training and experience in qualitative research methods. The transcribed texts were read repeatedly to achieve immersion in the data and to identify key phrases and meanings. Categories were developed based on the extracted content, and the data were coded accordingly. To enhance reflexivity, the researchers continuously reflected on their assumptions and potential biases throughout the analysis process. To strengthen coding reliability, emerging categories and themes were discussed among the research team and consensus was reached through iterative review and refinement. The coded data were then organized by domain, and overarching themes were derived through interpretation and synthesis.

## Ethics approval and consent to participate

This study was approved by the Sungshin Women's University Institutional Review Board (Approval Number: SSWUIRB-2023–025, 050). Informed written consent was obtained from all participants prior to their participation.

Participants were informed of their right to withdraw from the study at any time without penalty, and confidentiality was maintained throughout the study.

## Results

### Quantitative results

**Participants' general characteristics.** The mean age of the participants in the experimental group (n = 13) was 24.38 ± 4.93 years, while that in the control group (n = 12) was 22.67 ± 1.72 years. In the experimental group, most participants were fourth-year students (n = 10; 76.9%), whereas in the control group, most participants were third-year students (n = 7; 58.3%). Regarding self-perceived health status, 12 participants (92.3%) in the experimental group reported being "healthy," compared to 3 participants (25.0%) in the control group. Six participants in both experimental (46.2%) and control (50.0%) groups reported that they did regular exercises. Four participants (30.8%) in the experimental group reported that they followed a religion, compared to six participants (50.0%) in the control group (Table 3).

### Descriptive Statistics and Homogeneity Testing of Variables

Table 4 shows descriptive statistics and homogeneity test results for variables by group and time point. In the experimental group, mindfulness, gratitude disposition, and self-compassion scores increased in the post-test, whereas these scores of the participants in the control group decreased. Perceived stress of both groups decreased at post-test compared to pre-test, and mental health scores of both groups increased at post-test. To verify baseline homogeneity between groups, Mann–Whitney U tests were performed on pre-test scores of each variable. The results showed no statistically significant differences between groups for mindfulness (U = 115.50, p = .371) or self-compassion (U = 103.00, p = .767), which confirms the homogeneity of these variables. However, significant differences were found for gratitude disposition (U = 146.50, p = .019), perceived stress (U = 123.00, p = .017), and mental well-being (U = 138.50, p = .047). Baseline comparisons revealed significant group differences in gratitude disposition, perceived stress, and mental well-being. To account for these baseline variations and the small sample size, a nonparametric mixed-design approach using ART ANOVA was employed to examine group, time, and interaction effects.

### ART ANOVA by Group and Time

Interaction effects between group and time were found to be statistically significant for gratitude disposition ($F_{(1, 24.06)}$ = 11.34, p = .003), self-compassion ($F_{(1, 23.87)}$ = 15.62, p = .001), and perceived stress ($F_{(1, 24.16)}$ = 4.48, p = .045).

**Table 3. Participants' demographic characteristics.**

| Variable | Categories | n (%) or Mean ± SD | |
|---|---|---|---|
| | | Exp. (n = 13) | Cont. (n = 12) |
| Age (years) | | 24.38 ± 4.93 | 22.67 ± 1.72 |
| Year in university | 3rd | 3 (23.1) | 7 (58.3) |
| | 4th | 10 (76.9) | 5 (41.7) |
| Perceived health status | Unhealthy | 1 (7.7) | 6 (50) |
| | Fair | 0 (0) | 3 (25.0) |
| | Healthy | 12 (92.3) | 3 (25.0) |
| Regular exercise | Yes | 7 (53.8) | 6 (50.0) |
| | No | 6 (46.2) | 6 (50.0) |
| Religion | Yes | 4 (30.8) | 6 (50.0) |
| | No | 9 (69.2) | 6 (50.0) |

**Table 4. Descriptive statistics of variables and homogeneity test.**

| Variables | | Exp.(n=13) | Cont.(n=12) | U | p |
|---|---|---|---|---|---|
| | | M±SD | | | |
| Mindfulness | Pre-test | 59.08±7.19 | 61.17±6.77 | 115.50 | .371 |
| | Post-test | 62.46±10.49 | 57.83±6.59 | | |
| Gratitude disposition | Pre-test | 32.38±4.89 | 36.00±3.38 | 146.50 | .019 |
| | Post-test | 34.54±5.19 | 32.42±4.58 | | |
| Self-compassion | Pre-test | 39.54±8.60 | 40.92±6.65 | 103.00 | .767 |
| | Post-test | 42.31±8.66 | 36.67±5.03 | | |
| Perceived stress | Pre-test | 28.54±6.24 | 24.58±3.23 | 123.00 | .017 |
| | Post-test | 23.69±4.57 | 23.42±2.94 | | |
| Mental well-being | Pre-test | 36.62±11.59 | 43.58±9.35 | 138.50 | .047 |
| | Post-test | 42.77±8.68 | 46.92±6.64 | | |

(Table 5) To examine the simple main effects of the group factor following the ART ANOVA, Wilcoxon signed-rank tests were performed. The results indicated that changes in gratitude disposition were significant only in the control group (W = 0.00, p = .003). For self-compassion, statistically significant pre-post differences were observed in both the experimental (W = 65.00, p = .041) and control groups (W = 10.00, p = .040). Perceived stress showed a statistically significant change only in the experimental group (W = 13.00, p = .023) (Table 5; Fig 3, Fig 4, Fig 5). Effect size estimates ($\eta^2$) were calculated based on aligned rank sums of squares derived from the ART ANOVA. These values were uniformly small, reflecting the conservative nature of rank-based nonparametric methods. Because ART operates on aligned rank-transformed data instead of raw scores, variance-based indices such as $\eta^2$ are inherently attenuated, particularly in studies with small sample sizes or non-normal distributions. Accordingly, the observed effect sizes should be interpreted as a methodological characteristic of the analytic approach instead of as evidence of negligible intervention effects.

**Table 5. ART ANOVA for group and time.**

| Variables | | F | df1 | df2 | p | $\eta^2$ |
|---|---|---|---|---|---|---|
| Mindfulness | Group | 0.13 | 1 | 25.57 | .718 | .001 |
| | Time | 0.21 | 1 | 24.13 | .651 | |
| | Group × Time | 4.03 | 1 | 24.10 | .056 | |
| Gratitude disposition | Group | 0.57 | 1 | 25.61 | .457 | .002 |
| | Time | 0.07 | 1 | 24.01 | .792 | |
| | Group × Time | 11.34 | 1 | 24.06 | .003 | |
| Self-compassion | Group | 2.43 | 1 | 25.70 | .132 | .01 |
| | Time | 0.99 | 1 | 23.86 | .330 | |
| | Group × Time | 15.62 | 1 | 23.87 | .001 | |
| Perceived stress | Group | 5.05 | 1 | 25.38 | .034 | .000 |
| | Time | 14.57 | 1 | 24.18 | .001 | |
| | Group × Time | 4.48 | 1 | 24.16 | .045 | |
| Mental well-being | Group | 5.41 | 1 | 25.58 | .028 | .02 |
| | Time | 8.58 | 1 | 24.13 | .007 | |
| | Group × Time | 1.52 | 1 | 24.04 | .229 | |

df1 = degrees of freedom for effect; df2 = residual degrees of freedom.

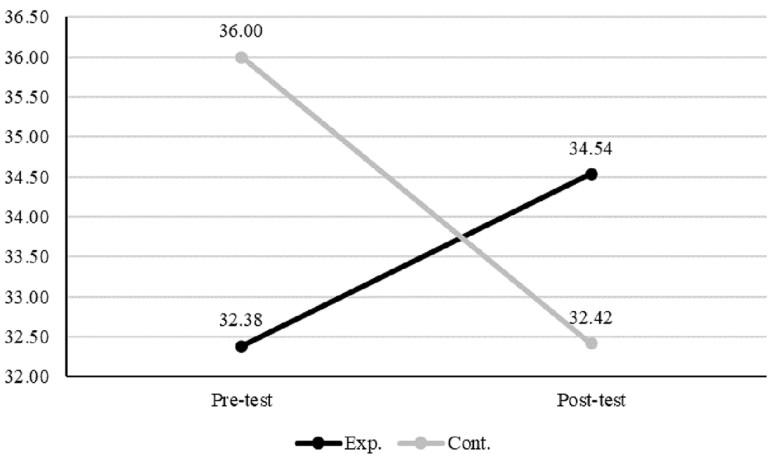

**Fig 3. Gratitude disposition.**

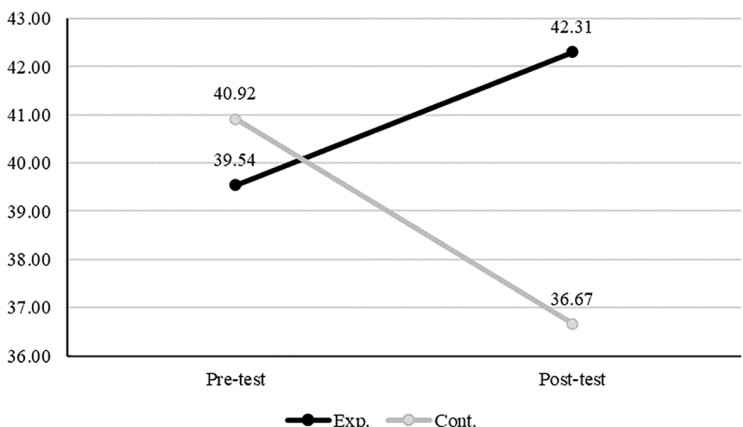

**Fig 4. Self-compassion.**

Confidence intervals were not reported, as conventional variance-based interval estimates are not statistically appropriate for rank-transformed data.

### Qualitative results

During the content analysis of the interview data, three overarching categories, each comprising nine subcategories were identified. These categories are used to describe experiences and perceived impacts of nursing students participating in the MMPT program. The categories were (A) the need for self-care and stress management, (B) development of positive psychological attitudes and personal growth, and (C) experiences during the program process. (Table 6)

**Category A: The need for self-care and stress management.** Participants commonly reported that they joined the program because of their growing awareness of the need to care for their well-being amid academic and clinical demands. In the subcategory, "awareness and practice of self-care," the participants shared their experience of having overlooked their mental health owing to their busy schedules and sought strategies to restore emotional balance. One participant

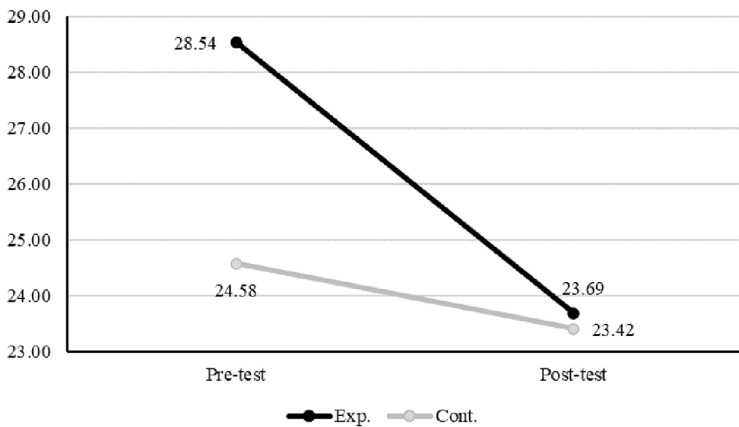

**Fig 5. Perceived stress.**

explained, 'I joined because I felt I could not take care of myself properly amidst my busy schedule.' Another noted, "Through the program, I realized the importance of caring for my mental health."

The subcategory, "balancing academics and personal life" reflected the participants' desire to simultaneously prioritize academic success and self-care. One participant stated, "I did not want to neglect my studies while trying to take care of myself." Another participant added, "I wanted to prioritize self-care while managing my academics."

**Category B: Development of positive psychological attitudes and personal growth.** The second major theme was centered on the participants' psychological and emotional development through the MMPT. Within the subcategory, "positive attitude and perspective change," the participants described their experience of a shift in how they viewed themselves and their daily lives. Several participants noted that keeping a journal of gratitude helped them reframe their everyday experiences more positively. As one participant said, "By writing a gratitude journal, I began to notice more positive aspects of daily life." Another reflected, "I thought participating in the program would make my job search feel less bleak."

The combination of breathing and self-compassion meditations was also found to contribute to emotional awareness and cognitive reframing. One participant said, "Through breathing meditation, I learned how to calm my mind and let go of distracting thoughts." Another remarked, "While practicing self-compassion meditation, I realized I had been too hard on myself and started to treat myself kindlier."

Under the subcategory of "coping with stress and emotional stability," descriptions of how students used meditation and mindfulness techniques to manage stress were included. The participants observed a greater sense of calmness and resilience. A participant said, "Even when I'm stressed, I do not get shaken up as I used to." Another observed, "Thanks to the habit of mindfulness and gratitude, I now feel more content and satisfied with my life."

Regarding the subcategory of "self-compassion and reflection," the participants discussed increased self-understanding and a kinder attitude toward themselves. The participants emphasized the value of having dedicated time for personal reflection each week.

Finally, the subcategory of "intention for continued practice" indicated that many participants intended to maintain mindfulness and gratitude practices after the program ended. For example, one participant stated, "I plan to keep practicing what I learned in the program."

**Category C: Experiences during the program process.** The third category encompassed students' initial responses to the program and their gradual adaptation. Regarding the subcategory of "initial awkwardness and challenges," several participants described the experience of having difficulty of concentration during early meditation sessions and discomfort

**Table 6. Summary of categories, subcategories, and meaning units regarding experiences in the MMPT Program.**

| Category | Subcategory | Meaning unit (example statement) |
|---|---|---|
| **A. The need for self-care and stress management** | Awareness and practice of self-care | "I joined because I felt I could not take care of myself properly amidst my busy schedule."; "Through the program, I realized how important it is to care for my mental health." |
| | Balancing academics and personal life | "I did not want to neglect my studies while also trying to take care of myself."; "I wanted to prioritize self-care while managing my academics as well." |
| **B. Development of positive psychological attitudes and personal growth** | Positive attitude and perspective change | "By writing a **gratitude** journal, I began to notice more positive aspects of daily life."; "I thought participating in the program would make my job search feel less bleak." "Through **breathing meditation**, I learned how to calm my mind and let go of distracting thoughts." "While practicing **self-compassion meditation**, I realized I had been too hard on myself and started to treat myself kindlier." |
| | Coping with stress and emotional stability | "I learned how to calm myself through meditation and breathing techniques."; "Even when I am stressed, I do not get shaken up as I used to." "Thanks to the habit of mindfulness and gratitude, I now feel more **content and satisfied** with my life." |
| | Self-understanding and reflection | "The program helped me understand myself better."; "I appreciated having time each week to reflect on myself." |
| | Intention for continued practice | "I plan to keep practicing what I learned in the program." |
| **C. Experiences during the program process** | Initial awkwardness and challenges | "At first, it was hard to concentrate on meditation, and writing the gratitude journal appeared awkward." |
| | Gradual adaptation and skill acquisition | "As the weeks passed, I **got used to meditation** and began to enjoy writing the gratitude journal." |
| | Empathy and support from peers | "As the weeks passed, I got used to meditation and began to enjoy writing the gratitude journal." |

with expressing gratitude. One participant remarked, "At first, it was hard to concentrate on meditation, and writing the gratitude journal felt awkward."

The subcategory "gradual adaptation and skill acquisition" captured how students became more comfortable with mindfulness practices. As one student observed, "As weeks passed, I got used to meditation and began to enjoy writing the gratitude journal."

Finally, in the subcategory, "empathy and support from peers," the participants described a sense of solidarity and emotional relief from sharing experiences with others. One participant said, "It was comforting to know others were struggling, too. We encouraged and supported one another."

These qualitative findings suggest that the MMPT program contributed to participants' enhanced emotional awareness, gratitude disposition, and coping strategies, while also fostering interpersonal empathy and self-compassion. The participants' narratives provided rich insights into the personal significance of the program and its relevance to nursing education.

## Discussion

This study aimed to evaluate the effectiveness of a brief MMPT program tailored for nursing students to reduce their stress and foster their psychological well-being. The program was designed based on needs assessment, intentionally shortened, and delivered online. The program was employed to ensure that its format and content were feasible and relevant to the unique demands of nursing students. Quantitative and qualitative data were analyzed with a convergent mixed-methods design to provide a comprehensive understanding of the program outcomes.

As this study was conducted as a small-scale pilot investigation, the sample size was limited and baseline differences between groups were observed. These characteristics reflect the exploratory nature of the study and warrant cautious interpretation of the quantitative findings. Within this context, quantitative findings demonstrated that the MMPT program demonstrated measurable benefits across several psychological domains, with distinct patterns of change in gratitude disposition, self-compassion, and perceived stress. While the control group exhibited a significant change in gratitude, the experimental group showed only a modest, nonsignificant improvement. Given the lower baseline scores of the participants in the experimental group, this result suggests a subtle positive effect of the intervention, although the magnitude of the change was insufficient for statistical confirmation. In contrast, self-compassion followed opposite trajectories between the groups; it increased in the experimental group but declined in the control group. This finding highlights the effectiveness of the program in cultivating more compassionate self-attitudes among the participants. This pattern indicates that the MMPT program may have functioned as a protective psychological resource, buffering participants against self-critical responses commonly elicited by academic and clinical stress. From a positive psychology perspective, self-compassion operates as an intrapersonal resource that mitigates self-criticism and supports adaptive emotional coping under conditions of chronic stress [17,41]. The observed increase indicates that compassion-based practices embedded in the MMPT program may have facilitated a more supportive internal dialogue when participants encountered academic and clinical challenges. Perceived stress also decreased significantly in the experimental group, implying that the intervention influenced not only stress exposure but also stress appraisal and regulation processes. Consistent with stress appraisal theory, changes in perceived stress reflect shifts in how individuals evaluate and respond to stressors instead of changes in stressors themselves [42]. From a mindfulness-based perspective, enhanced present-moment awareness and nonjudgmental attention may reduce cognitive reactivity, allowing individuals to respond with greater emotional flexibility instead of habitual rumination or avoidance [43,44].

As previously mentioned, quantitative findings indicated that the MMPT program was associated with significant improvements in self-compassion and perceived stress, along with a positive trend in gratitude disposition. These effects were further illuminated by the qualitative findings, particularly within the theme of development of positive psychological attitudes and personal growth. Participants described increased self-kindness, emotional stability, and more adaptive perspectives toward daily stressors, which correspond closely with the observed quantitative gains in self-compassion and stress reduction. From a positive psychology perspective, self-compassion functions as a core intrapersonal resource that buffers stress by promoting emotional acceptance and reducing self-criticism [17,36]. Participants' narratives describing kinder self-attitudes and improved emotional regulation provide experiential support for this mechanism. Similarly, gratitude journaling facilitated positive cognitive reappraisal, enabling participants to recognize meaningful aspects of everyday experiences. This qualitative pattern aligns with prior evidence that gratitude-based and self-affirmative practices enhance psychological well-being by reinforcing positive self-representations and emotional balance [18,19,33]. The integration of mindfulness with positive psychology appears to be particularly relevant in explaining why reductions in perceived stress emerged quantitatively, even when changes in mindfulness scores were not immediately significant. Mindfulness practices may have supported awareness and emotional regulation, while positive psychology components—such as gratitude and self-compassion—actively fostered adaptive meaning-making and motivation, consistent with integrative models of well-being [16,20]. Qualitative reports of calmness, increased positivity, and sustained coping efforts indicate that these mechanisms were subjectively internalized, even though not yet fully captured by standardized mindfulness measures.

By contrast, mindfulness and overall mental health did not demonstrate significant immediate changes. This may be explained by the nature of these constructs, which often requires extended practice or longitudinal reinforcement before robust effects emerge. Previous research has shown that mindfulness-related improvements are more frequent at follow-up than immediately after an intervention [9,45]. Mindfulness is not merely a skill acquired through brief exposure but a dispositional capacity that develops gradually through sustained and repeated practice [45]. In short-term interventions, participants may begin to internalize mindfulness-related attitudes—such as present-moment awareness and nonjudgmental acceptance—before these changes are fully captured by self-report measures. Similarly, mental well-being represents a broad, multidimensional outcome encompassing emotional, psychological, and social functioning [29]. Improvements in such global indicators often comprise downstream effects of more proximal psychological resources, including self-compassion and adaptive stress regulation. In this study, qualitative findings indicated early experiential shifts—such as increased calmness, emotional awareness, and positive reframing—that may precede measurable changes in overall mental well-being. Prior research indicates that these foundational changes often emerge first, with broader well-being outcomes becoming evident only after longer follow-up periods or continued practice [10,45]. Additionally, qualitative accounts of increased awareness and emotional regulation imply that early experiential changes may precede detectable quantitative shifts. Another possible explanation is the relatively small sample size, which limited the statistical power to detect subtler effects [39]. Taken together, these findings suggest that the MMPT program effectively enhanced self-compassion and reduced stress in the short term, with preliminary indications of positive changes in gratitude disposition. However, outcomes, such as mindfulness and mental health, may require more time, practice, or larger-scale interventions to provide more substantial benefits.

In addition to psychological outcomes, qualitative findings from categories A and C provide important contextual insights into how and why the intervention was experienced as meaningful. Category A ("the need for self-care and stress management") highlights that participants entered the program with an explicit awareness of unmet self-care needs, implying that MMPT addressed a perceived gap in existing nursing education instead of introducing an externally imposed intervention. This readiness for change may have facilitated engagement and receptivity to the program content. Category C ("experiences during the program process") illustrates the dynamic nature of skill acquisition, with initial discomfort giving way to gradual adaptation and increased confidence in mindfulness and gratitude practices. Peer empathy and shared reflection emerged as key facilitators, reinforcing motivation and normalizing stress experiences. Together, these findings indicate that MMPT's effectiveness may depend not only on its psychological techniques but also on its process-oriented and relational components, which support sustained engagement and meaning-making.

A key strength of this study was its foundation of needs assessment, which ensured that the program content addressed the specific challenges faced by nursing students. Barriers to participation were minimized through the shortening of the duration, reduced session length, and online delivery of the program. The inclusion of structured assignments and group interactions promotes consistent engagement. These design elements enhanced feasibility and demonstrated how well-being programs can be realistically integrated into the demanding academic schedules of nursing students. Furthermore, the use of a convergent mixed-methods design strengthened the evaluation by capturing measurable outcomes and rich personal experiences.

Building on these findings, future research should further examine the mechanisms through which integrated mindfulness–positive psychology interventions exert their effects, particularly the roles of self-compassion and stress appraisal processes. The present results indicate that compassion-based practices may function as key intrapersonal mechanisms by reducing self-criticism and fostering adaptive emotional regulation under academic and clinical stress. From a mindfulness perspective, improvements in stress outcomes may reflect changes in cognitive reactivity and appraisal instead of reductions in stress exposure per se [42,43]. In the context of Korean nursing education, where academic pressure, performance-oriented evaluation, and self-critical norms are prevalent, interventions that explicitly cultivate self-kindness and emotional acceptance may hold particular relevance [1,16]. Therefore, brief, structured, and online-delivered

programs such as MMPT may offer a culturally responsive and feasible approach to supporting nursing students' psychological well-being within constrained educational settings. Future studies should further examine culturally embedded mechanisms of change and evaluate how such interventions can be integrated into nursing curricula through longitudinal and multi-institutional designs.

This study has a few limitations that should be considered when interpreting the findings. First, the study employed a pilot design with a relatively small sample size drawn from a limited number of institutions, which restricts the generalizability of the results. As a pilot study, the sample size was intentionally limited, and this constraint was also reflected in the study title. Second, participants were recruited through online announcements using convenience sampling, which may have introduced self-selection bias, as students with a greater interest in mindfulness or psychological well-being may have been more inclined to participate. Third, the absence of long-term follow-up precludes conclusions about the sustainability of program effects. Reliance on self-reported questionnaires introduces potential bias, and unmeasured external influences may have affected the outcomes of the control group. Fourth, although recorded sessions were provided only when live attendance was not feasible, the use of two delivery modalities (live and recorded) may have introduced variability in participant engagement and intervention fidelity. Finally, the intervention was facilitated by the authors, which may have introduced experimenter bias and should be considered when interpreting the findings. These limitations highlight the need for future controlled trials with larger and more diverse populations and extended follow-up assessments.

This pilot study provides preliminary evidence that a brief needs-based MMPT program can be effective for reducing stress and fostering positive psychological growth in nursing students. Quantitative analyses employed in the study demonstrated significant improvements in participants' gratitude, self-compassion, and stress reduction, whereas qualitative findings revealed broader benefits across all domains, including mindfulness and mental health. The integration of these results underscores the value of mixed-method designs in capturing both measurable and experiential impacts of interventions. As the program comprised a shortened version, delivered online, and adapted to student needs, it represented a practical and innovative model for supporting well-being in nursing education. With further refinement, larger-scale testing, and cultural adaptation, such interventions hold promise for improving both students' mental health and the resilience of the future nursing workforce.

## Supporting information

**S1 Data. Minimal dataset.** Contains raw quantitative survey data used in all statistical analyses (pre- and post-test scores for mindfulness, perceived stress, gratitude disposition, self-compassion, and mental well-being).
(XLSX)

**S2 Data. Interview excerpts.** Includes de-identified qualitative excerpts used in the inductive content analysis, organized by category (A–C).
(DOCX)

## Author contributions

**Conceptualization:** Young Im Cho, Dong Hee Kim.

**Data curation:** Young Im Cho, Hyo Jin Kim.

**Formal analysis:** Hyo Jin Kim, Dong Hee Kim.

**Funding acquisition:** Dong Hee Kim.

**Investigation:** Young Im Cho, Dong Hee Kim.

**Methodology:** Young Im Cho, Dong Hee Kim.

**Project administration:** Dong Hee Kim.

**Supervision:** Dong Hee Kim.

**Validation:** Young Im Cho, Hyo Jin Kim, Dong Hee Kim.

**Writing – original draft:** Young Im Cho, Hyo Jin Kim, Dong Hee Kim.

**Writing – review & editing:** Young Im Cho, Hyo Jin Kim, Dong Hee Kim.

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
