## [Decision Letter · Decision Letter 0]

4 Dec 2025

Dear Dr. Kim,

Thank you for submitting your manuscript to PLOS ONE. After careful consideration, we feel that it has merit but does not fully meet PLOS ONE’s publication criteria as it currently stands. Therefore, we invite you to submit a revised version of the manuscript that addresses the points raised during the review process.

I fully agree with the revisions requested by the first reviewer, especially with regard to the methodology section, which needs to be improved. Greater attention should also be paid to the limitations of the article and possible future developments.

We look forward to receiving your revised manuscript.

Kind regards,

Ramona Bongelli, Ph.D.

Academic Editor

PLOS ONE

Journal Requirements:

3. We note that your Data Availability Statement is currently as follows: “All relevant data are within the manuscript and its Supporting Information files.”

Please confirm at this time whether or not your submission contains all raw data required to replicate the results of your study. Authors must share the “minimal data set” for their submission. PLOS defines the minimal data set to consist of the data required to replicate all study findings reported in the article, as well as related metadata and methods (https://journals.plos.org/plosone/s/data-availability#loc-minimal-data-set-definition ).).

Additional Editor Comments :

Dear authors,

Thank you for submitting your interesting article.

I fully agree with the revisions requested by the first reviewer, especially with regard to the methodology section, which needs to be improved. Greater attention should also be paid to the limitations of the article and possible future developments.

Reviewers' comments:

Reviewer's Responses to Questions

**Comments to the Author**

1. Is the manuscript technically sound, and do the data support the conclusions?

Reviewer #1: Yes

Reviewer #2: Yes

2. Has the statistical analysis been performed appropriately and rigorously?

Reviewer #1: Yes

Reviewer #2: Yes

3. Have the authors made all data underlying the findings in their manuscript fully available?

Reviewer #1: Yes

Reviewer #2: Yes

4. Is the manuscript presented in an intelligible fashion and written in standard English?

Reviewer #1: Yes

Reviewer #2: Yes

Reviewer #1: In this study, a modified mindfulness, meditation, and positive psychology training program (MMPT) was piloted. The curriculum was created with nursing students' needs in mind, making it shorter and conducted online. My observations are given below:

(1) The study title is okay.

(2) The abstract needs minor revision. The abstract successfully conveys both quantitative and qualitative findings and is organized and easy to read. However, it lacks a clear statement of objectives, overemphasizes limitations, and is excessively detailed for an abstract. It is advised to pay more attention to the theoretical justification and implications.

(3) The introduction also needs minor revision. It is thorough and organized, providing a concise justification and background information for the research. It logically leads to the necessity of an integrated mindfulness–positive psychology intervention and successfully draws attention to the issue of stress among nursing students. But it is a little long and could be shorter by omitting the repetition of the negative effects of stress. More seamless linkage is required when moving from general stress literature to the MMPT program. Furthermore, to support the research's novelty and justification, the gap in the literature and the study's unique contribution need to be more clearly articulated. Please include research that describes the effectiveness of other similar positive psychotherapies such as self-affirmation and self-compassion and uncovers their mechanism based on the following research:

Tiwari, G. K., Choudhary, A., Singh, A., Shukla, A., Macorya, A. K., Pandey, A., & Singh, A. K. (2025). Enhancing Self‐Esteem: Evaluating the Effects of a Self‐Affirmation Intervention Among Indian Adults With Subclinical Depression. Counselling and Psychotherapy Research, 25(1), e12892. https://doi.org/10.1002/capr.12892

Tiwari, G. K., Pandey, R., Rai, P. K., Pandey, R., Verma, Y., Parihar, P., Ahirwar, G., Tiwari, A. S., & Mandal, S. P. (2020). Self-compassion as an intrapersonal resource of perceived positive mental health outcomes: A thematic analysis. Mental Health, Religion & Culture, 23(7), 550–569. https://doi.org/10.1080/13674676.2020.1774524

Tiwari, G. K., Parihar, P., Singh, A. K., Macorya, A. K., Shukla, A., Singh, A., Choudhary, A., & Pandey, A. (2025). Understanding the nature and dynamics of self-affirmation in non-depressed and subclinically depressed Indian adults: A thematic analysis. BMC Psychiatry, 25(1), 224. https://doi.org/10.1186/s12888-024-06364-0

(4) The methods section also needs improvements. Although the methodology is ethically sound and generally well-structured, its rigor is undermined by a number of limitations. The statistical power and generalizability are significantly restricted by the small sample size (n=25). Online recruitment for convenience sampling may result in self-selection bias. A more convincing argument for the superiority of ART-ANOVA over more straightforward techniques is required, even though it is suitable for nonparametric data. Treatment fidelity may be compromised by the intervention's two delivery modalities (live and recorded). Additionally, the program was facilitated by the authors themselves, which could have introduced experimenter bias. Information about researcher reflexivity and coding reliability is absent from qualitative methods.

(5) The results section also needs revision. Interpretability is limited by certain issues, but the results section clearly presents both quantitative and qualitative findings. External validity and statistical power are diminished by the small sample size (n=25). Group equivalence is jeopardized by baseline variations in mental health, perceived stress, and gratitude, which makes it challenging to fully credit the intervention for post-test gains. ART ANOVA and suitable nonparametric tests were employed, but effect sizes and confidence intervals are absent. The overall triangulation and interpretive depth are weakened by the inadequate integration of qualitative findings with quantitative results, despite the fact that the qualitative findings are rich and well-structured into cohesive themes.

(6) The discussion section also needs revision. Overall, it is well-structured and offers a fair combination of qualitative and quantitative data. Nevertheless, it frequently lacks critical interpretation of data set inconsistencies and is unduly descriptive. Deeper theoretical engagement with frameworks from positive psychology and mindfulness to explain observed outcomes would enhance the discussion. Rather than using conceptual reasoning, the explanation for non-significant results mainly depends on sample size. Additionally, the implications for future research and practice are presented in a general way without offering clear, practical guidance. This section would be strengthened by a stronger focus on mechanism-based interpretations and cultural contextualization.

(7) The references are okay. Please correct it by following the journal’s guidelines. The tables and figures should be prepared following standard guidelines.

Reviewer #2: The manuscript is well structured and accurately explains the studies that were carried out to reach the above conclusions. The study is very interesting and contributes innovatively to research. The main limitation is the small sample size that was used and the reduction of the study (hours, questions, etc.). Considering the results obtained, it might be interesting to extend and expand the study.

**Do you want your identity to be public for this peer review?** For information about this choice, including consent withdrawal, please see our For information about this choice, including consent withdrawal, please see our Privacy Policy .

Reviewer #1: **Yes:** Gyanesh Kumar TiwariGyanesh Kumar Tiwari

Reviewer #2: No

You may also use PLOS’s free figure tool, NAAS, to help you prepare publication quality figures: https://journals.plos.org/plosone/s/figures#loc-tools-for-figure-preparation

---

## [Author Response · Author response to Decision Letter 1]

27 Jan 2026

RESPONSES TO REVIEWERS’ COMMENTS

PONE-D-25-49792

Title: Pilot study on the effect of a Mindfulness–Meditation–Positive Psychology Training program on perceived stress and mental well-being in Korean nursing students: A mixed methods analysis

Journal Requirements:

If applicable, we recommend that you deposit your laboratory protocols in protocols.io to enhance the reproducibility of your results.

Authors’ Response ⇒ This study did not involve any laboratory procedures; therefore, there are no protocols applicable for deposition in protocols.io.

Authors’ Response ⇒ We have reviewed the PLOS ONE style requirements, including the file naming guidelines, and have ensured that the manuscript adheres to all relevant formatting guidelines.

Authors’ Response ⇒ The ethics statement is included only in the Methods section.

3. We note that your Data Availability Statement is currently as follows: “All relevant data are within the manuscript and its Supporting Information files.”

Authors’ Response ⇒ All raw data required to replicate the study findings, including survey and interview data, have been provided in their entirety in the manuscript and Supporting Information files.

Authors’ Response ⇒ The recommended references were reviewed and included, as they were deemed relevant to the study.

Authors’ Response ⇒ The reference list was carefully reviewed to ensure completeness and accuracy. No retracted articles were cited in the manuscript; therefore, no changes were required.

Additional Editor Comments :

Dear authors,

Thank you for submitting your interesting article.

I fully agree with the revisions requested by the first reviewer, especially with regard to the methodology section, which needs to be improved. Greater attention should also be paid to the limitations of the article and possible future developments.

Authors’ Response ⇒ We carefully reviewed all comments of the first reviewer. As comments (1)–(4) and (6) were considered acceptable and required no revisions, we have provide below a focused response addressing Comment (5), which concerns methodological rigor, limitations, and future directions.

5. Review Comments to the Author

Reviewer #1: In this study, a modified mindfulness, meditation, and positive psychology training program (MMPT) was piloted. The curriculum was created with nursing students' needs in mind, making it shorter and conducted online. My observations are given below:

(1) The study title is okay.

Authors’ Response ⇒ Thank you.

(2) The abstract needs minor revision. The abstract successfully conveys both quantitative and qualitative findings and is organized and easy to read. However, it lacks a clear statement of objectives, overemphasizes limitations, and is excessively detailed for an abstract. It is advised to pay more attention to the theoretical justification and implications.

Authors’ Response ⇒ The abstract was revised to clarify the study objective, reduce emphasis on limitations, and streamline content. In addition, the theoretical rationale and practical implications of integrating mindfulness and positive psychology were strengthened; further, quantitative and qualitative findings were better aligned.

(3) The introduction also needs minor revision. It is thorough and organized, providing a concise justification and background information for the research. It logically leads to the necessity of an integrated mindfulness–positive psychology intervention and successfully draws attention to the issue of stress among nursing students.

But it is a little long and could be shorter by omitting the repetition of the negative effects of stress.

Authors’ Response ⇒ Thank you for your kind comment! Repetitive descriptions of the negative effects of stress were removed from the Introduction to improve conciseness while retaining the core rationale for the study. (lines 51–54)

Furthermore, to support the research's novelty and justification, the gap in the literature and the study's unique contribution need to be more clearly articulated.

Authors’ Response ⇒ As per recommendation, we have clarified the research gap and highlighted the need for a brief, integrated mindfulness–positive psychology intervention for nursing students. (lines 96–99)

More seamless linkage is required when moving from general stress literature to the MMPT program.

Authors’ Response ⇒ As per instruction, we have improved the linkage between the general stress literature and the MMPT framework by restructuring the Introduction to articulate the need for stress management education, followed by evidence supporting mindfulness-based interventions, and subsequently their practical and theoretical limitations, thereby providing a clearer rationale for the MMPT program. (lines 55–95)

Please include research that describes the effectiveness of other similar positive psychotherapies such as self-affirmation and self-compassion and uncovers their mechanism based on the following research:

Tiwari, G. K., Choudhary, A., Singh, A., Shukla, A., Macorya, A. K., Pandey, A., & Singh, A. K. (2025). Enhancing Self‐Esteem: Evaluating the Effects of a Self‐Affirmation Intervention Among Indian Adults With Subclinical Depression. Counselling and Psychotherapy Research, 25(1), e12892. https://doi.org/10.1002/capr.12892

Tiwari, G. K., Pandey, R., Rai, P. K., Pandey, R., Verma, Y., Parihar, P., Ahirwar, G., Tiwari, A. S., & Mandal, S. P. (2020). Self-compassion as an intrapersonal resource of perceived positive mental health outcomes: A thematic analysis. Mental Health, Religion & Culture, 23(7), 550–569. https://doi.org/10.1080/13674676.2020.1774524

Tiwari, G. K., Parihar, P., Singh, A. K., Macorya, A. K., Shukla, A., Singh, A., Choudhary, A., & Pandey, A. (2025). Understanding the nature and dynamics of self-affirmation in non-depressed and subclinically depressed Indian adults: A thematic analysis. BMC Psychiatry, 25(1), 224. https://doi.org/10.1186/s12888-024-06364-0

Authors’ Response ⇒ The suggested studies on self-affirmation and self-compassion have been added to strengthen the theoretical justification. (lines 76-81, 421, 440, 446)

(4) The methods section also needs improvements. Although the methodology is ethically sound and generally well-structured, its rigor is undermined by a number of limitations. The statistical power and generalizability are significantly restricted by the small sample size (n=25). Online recruitment for convenience sampling may result in self-selection bias.

Authors’ Response ⇒ We acknowledge the reviewer’s concern regarding the small sample size and sampling method. This study was designed as a pilot study, which has been clearly indicated in the title; the limited sample size was intended to assess feasibility and preliminary effects instead of generalizability. The issue of restricted statistical power due to the small sample size has been explicitly addressed in the limitations in the Discussion section. In addition, in the penultimate paragraph of the Discussion section, we have highlighted the limitations to further acknowledge the use of online convenience sampling and the potential risk of self-selection bias. These points have been clearly discussed to ensure transparency and to guide the design of future large-scale studies. (lines 404–407, 519–525)

A more convincing argument for the superiority of ART-ANOVA over more straightforward techniques is required, even though it is suitable for nonparametric data.

Authors’ Response ⇒ We have added a concise rationale for using ART ANOVA, highlighting its suitability for analyzing interaction effects in studies employing the repeated-measures design. (lines 245–247)

Treatment fidelity may be compromised by the intervention's two delivery modalities (live and recorded).

Authors’ Response ⇒ We appreciate the reviewer’s thoughtful comment regarding potential treatment fidelity concerns related to the use of both live and recorded sessions. Recorded sessions were provided only to participants who were unavoidably absent (e.g., health issues) to ensure equitable access to the intervention. To maintain intervention fidelity, all recordings followed the same standardized session structure and content as the live sessions, and participants were required to complete the same practice assignments. This limitation has been acknowledged in the Methods and Discussion sections. (lines 134–136, 528–530)

Additionally, the program was facilitated by the authors themselves, which could have introduced experimenter bias.

Authors’ Response ⇒ Potential experimenter bias due to authors’ involvement has been acknowledged in the Discussion section where limitations are delineated. (lines 530–532)

Information about researcher reflexivity and coding reliability is absent from qualitative methods.

Authors’ Response ⇒ Thank you for this comment. We have revised the qualitative methods section to include information on researcher reflexivity and coding procedures. (lines 252–261)

(5) The results section also needs revision. Interpretability is limited by certain issues, but the results section clearly presents both quantitative and qualitative findings. External validity and statistical power are diminished by the small sample size (n=25). Group equivalence is jeopardized by baseline variations in mental health, perceived stress, and gratitude, which makes it challenging to fully credit the intervention for post-test gains.

Authors’ Response ⇒ The limited external validity and statistical power associated with the small sample size were addressed in the study design as a pilot investigation and are explicitly discussed in the Discussion section. (lines 401–404, 517–522) In addition, to address baseline differences, ART ANOVA was applied to examine group-by-time interaction effects; this analytical rationale has been clarified in the Results section. (lines 297–301)

ART ANOVA and suitable nonparametric tests were employed, but effect sizes and confidence intervals are absent.

Authors’ Response ⇒ Thank you for this helpful suggestion. In response, we have added effect size estimates for the ART ANOVA results to enhance the transparency and interpretability of the findings (Table 5). Notably, ART ANOVA is a rank-based nonparametric method, and effect size estimates derived from rank-transformed data tend to be smaller because information about the original scale variance is attenuated during the transformation process. Accordingly, these effect sizes should be interpreted as reflecting methodological characteristics of the ART approach instead of as indications of trivial effects. Confidence intervals were not calculated for the ART-based effect sizes, as conventional variance-based confidence interval estimation relies on distributional assumptions that are not directly applicable to rank-transformed data.

The overall triangulation and interpretive depth are weakened by the inadequate integration of qualitative findings with quantitative results, despite the fact that the qualitative findings are rich and well-structured into cohesive themes.

Authors’ Response ⇒ We sincerely appreciate this critical comment. In response, we revised the Discussion section to strengthen the integration of quantitative and qualitative findings. (lines 417–454)

(6) The discussion section also needs revision.

Overall, it is well-structured and offers a fair combination of qualitative and quantitative data.

Authors’ Response ⇒ Thank you.

Nevertheless, it frequently lacks critical interpretation of data set inconsistencies and is unduly descriptive. Deeper theoretical engagement with frameworks from positive psychology and mindfulness to explain observed outcomes would enhance the discussion. Rather than using conceptual reasoning, the explanation for non-significant results mainly depends on sample size. Additionally, the implications for future research and practice are presented in a general way without offering clear, practical guidance. This section would be strengthened by a stronger focus on mechanism-based interpretations and cultural contextualization.

Authors’ Response ⇒ In response to the reviewer’s comments, we have substantially revised the Discussion to enhance theoretical depth, mixed-methods integration, and interpretive rigor. Quantitative and qualitative findings are now more explicitly connected using mindfulness and positive psychology frameworks, and non-significant results are discussed through conceptual and process-based lenses instead of methodological limitations alone. We have also refined the implications section to provide clearer, culturally informed guidance for future research and practice. (lines 417–454, 459–473, 503–517)

(7) The references are okay. Please correct it by following the journal’s guidelines. The tables and figures should be prepared following standard guidelines.

Authors’ Response ⇒ Thank you for your comment. All references were reviewed and formatted in accordance with the journal’s guidelines, and all tables and figures were revised to comply with the standard formatting requirements.

Reviewer #2: The manuscript is well structured and accurately explains the studies that were carried out to reach the above conclusions. The study is very interesting and contributes innovatively to research. The main limitation is the small sample size that was used and the reduction of the study (hours, questions, et

---

## [Decision Letter · Decision Letter 1]

2 Mar 2026

Pilot study on the effect of a Mindfulness-Meditation-Positive-Psychology-Training program on perceived stress and mental well-being in Korean nursing students: A mixed methods analysis

PONE-D-25-49792R1

Dear Dr. Kim,

We’re pleased to inform you that your manuscript has been judged scientifically suitable for publication and will be formally accepted for publication once it meets all outstanding technical requirements.

Kind regards,

Ramona Bongelli, Ph.D.

Academic Editor

PLOS One

Additional Editor Comments (optional):

After a careful review of the revised manuscript and the detailed point-by-point responses to the reviewers’ comments, I am pleased to state that the paper is now suitable for publication. The authors have addressed all concerns in a comprehensive and rigorous manner, substantially strengthening the theoretical framework, methodological transparency, statistical justification, and integration of their findings. Overall, the manuscript has been significantly refined and is, in my view, worthy of acceptance for publication.

Reviewers' comments:

Reviewer's Responses to Questions

**Comments to the Author**

Reviewer #1: All comments have been addressed

2. Is the manuscript technically sound, and do the data support the conclusions?

Reviewer #1: Yes

3. Has the statistical analysis been performed appropriately and rigorously?

Reviewer #1: Yes

4. Have the authors made all data underlying the findings in their manuscript fully available?

Reviewer #1: Yes

5. Is the manuscript presented in an intelligible fashion and written in standard English?

Reviewer #1: Yes

Reviewer #1: I carefully read the revised manuscript and found that the author/s have satisfactorily addressed all thew issues raised during the last submission. I am confident that the manuscript in its current form may contribute significantly to the literature.

**Do you want your identity to be public for this peer review?** For information about this choice, including consent withdrawal, please see our For information about this choice, including consent withdrawal, please see our Privacy Policy .

Reviewer #1: **Yes:** Gyanesh Kumar TiwariGyanesh Kumar Tiwari

---

## [Editor Report · Acceptance letter]

PONE-D-25-49792R1

PLOS One

Dear Dr. Kim,

I'm pleased to inform you that your manuscript has been deemed suitable for publication in PLOS One. Congratulations! Your manuscript is now being handed over to our production team.

Kind regards,

on behalf of

Professor Ramona Bongelli

Academic Editor

PLOS One